# LATENTPOISON – ADVERSARIAL ATTACKS ON THE LATENT SPACE

## ABSTRACT

Robustness and security of machine learning (ML) systems are intertwined, wherein a non-robust ML system (classifiers, regressors, etc.) can be subject to attacks using a wide variety of exploits. With the advent of scalable deep learning methodologies, a lot of emphasis has been put on the robustness of supervised, unsupervised and reinforcement learning algorithms. Here, we study the robustness of the latent space of a deep variational autoencoder (dVAE), an unsupervised generative framework, to show that it is indeed possible to perturb the latent space, flip the class predictions and keep the classification probability approximately equal before and after an attack. This means that an agent that looks at the outputs of a decoder would remain oblivious to an attack.

## 1 INTRODUCTION

The ability to encode data reliably is essential for many tasks including image compression, data retrieval and communication. As data is transmitted between communication channels, error detection and correction is often employed to deduce the presence of erroneous bits (Peterson & Weldon, 1972). The source of such errors can be a result of imperfection in the transmitter, channel or in the receiver. Often times, such errors can be deliberate where a man-in-middle attack (Desmedt, 2011; Conti et al., 2016) can result in deleterious erasure of information, yet to the receiver, it may end up as appearing untampered (Kos et al., 2017).

In deep learning, we are able to learn an encoding process using unsupervised learning such as in autoencoders (AE) (Kingma & Welling, 2013); however, we are less able to design methods for checking whether encodings have been tampered with. Therefore, there are two facets of this problem – the first, is to come up with methodologies of tampering with the models and second, is to detect the adversarial breach. In what follows, we will concentrate only on the first problem by presenting a method for tampering autoencoders. An autoencoder has two components: the encoder maps the input to a latent space, while the decoder maps the latent space to the requisite output. A vanilla autoencoder can, therefore, be used to compress the input to a lower dimensional latent (or feature) space. Other forms of autoencoder include the denoising AE (Vincent et al., 2010) that recovers an undistorted input from a partially corrupted input; the compressive AE (Theis et al., 2017) designed for image compression and the variational AE (Kingma & Welling, 2013) that assumes that the data is generated from a directed graphical model with the encoder operationalized to learn the posterior distribution of the latent space. Autoencoders have wide use in data analytics, computer vision, natural language processing, etc.

We propose an attack that targets the latent encodings of autoencoders, such that if an attack is successful the output of an autoencoder will have a different semantic meaning to the input. Formally, we consider an autoencoder consisting of an encoder and decoder model designed to reconstruct an input data sample such that the label information associated with the input data is maintained. For example, consider a dataset of images, $x$ with the labels, $y = \{0, 1\}$, and an encoder, $E : x \rightarrow z$ and a decoder, $D : z \rightarrow x$ where $z$ is a latent encoding for $x$. If the encoder and decoder are operating normally, the label of the reconstructed data sample, $\hat{y} = class(D(E(x)))$ should be the same as the label of the input data sample, where $class(\cdot)$ is the *soft* output of a binary classifier.

In this paper, we focus on learning an *attack* transformation, $\mathcal{T} \circ z$, such that if $z$ is the latent encoding for a data sample, $x$, with label 0, $\mathcal{T} \circ z$ is the latent encoding for a data sample with label 1. The

attack is designed to *flip* the label of the original input and change its content. Note that the same $\mathcal{T}$ is applied to each encoding and is not specific to either the input data sample or the encoding, it is only dependent on the label of the input data sample.

The success of an attack may be measured in three ways:

1. The number of elements in the latent encoding, changed by the attack process should be small. If the encoding has a particular length, changing multiple elements may make the attack more detectable.

2. When a decoder is applied to tampered encodings, the decoded data samples should be indistinguishable from other decoded data samples that have not been tampered with.

3. Decoded tampered-encodings should be classified with opposite label to the original (un-tampered) data sample.

Our contribution lies in studying transforms with these properties. Experimentally, we find that optimizing for requirement (1) may implicitly encourage requirement (2). Crucially, in contrast to previous work (Goodfellow et al., 2014), our approach does not require knowledge of the model (here a VAE) parameters; we need access only to the encodings and the output of a classifier, making our approach more practical (Papernot et al., 2017). Finally, we owe the success of this attack method primarily to the near-linear structure of the VAE latent space (Kingma & Welling, 2013) – which our attack exploits.

## 2 Comparison to Previous Work

Security in deep learning algorithms is an emerging area of research. Much focus has gone into the construction of adversarial data examples, inputs that are modified such that they cause deep learning algorithms to fail. Previous work, designing adversarial images, has focused on perturbing input data samples such that a classifier miss classifies adversarial examples (Goodfellow et al., 2014). The perturbation is intended to be so small that a human cannot detect the difference between the original data samples, and its adversarial version. Goodfellow et al. (Goodfellow et al., 2014) propose adding a perturbation proportional to $\text{sign}(\nabla_x J(\theta, x, y))$ where $J$ is the cost function used to train a classifier (that is being attacked), $\theta$ are the parameters of that classifier, and $x$ and $y$ the data and label pair, respectively. This type of attack requires the attacker to have high-level access to the classifiers' parameters and cost function. An alternative approach that does not require the adversary to have access to the model parameters, is presented by Papernot et al. (Papernot et al., 2017) who propose a more practical approach, requiring only the classifier output and knowledge of the encoding size. Our adversary has similar, practical requirements.

Our approach, is thus tangential to the previous work on adversarial images for classification. We focus on a man-in-middle form of attack (Diffie & Hellman, 1976): rather than launching an attack on data samples, we launch an attack on an intermediate encoding such that a *message* being sent from a sender is different to the *message* received by a receiver. Similar to previous work, we do not want the attack on the encoding to be detectable, but in contrast to previous work (Goodfellow et al., 2014; Papernot et al., 2017), we wish for the *message* – in this example the images – to be detectably changed, while still being consistent with other non-tampered *messages*.

Our work is more similar to that of Kos et al. (Kos et al., 2017) – in the sense that they propose attacking variational autoencoders in a similar sender-receiver framework. Their goal is to perform an attack on inputs to an autoencoder such that output of the autoencoder belongs to a different class to the input. For example, an image of the digit 8 is encoded, but following an attack, the decoded image is of the digit 7 (Kos et al., 2017). While the overall goal is very similar, their approach is very different since they focus on perturbing images – while we perturb latent encodings. This difference is illustrated in Figure 1.

Finally, most previous work (Goodfellow et al., 2014; Papernot et al., 2017; Kos et al., 2017) requires the calculation of a different perturbation for each adversarial example. Rather, in our approach, we learn a single (additive) adversarial perturbation that may be applied to almost any encoding to launch a successful attack. This makes our approach more practical for larger scale attacks.

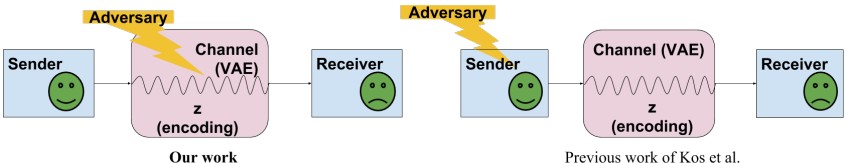

Figure 1: **Comparison of our work to previous work**. In both works, the sender sends an image belonging to one class and the receiver receives an image belonging to a different class, "smile" or "no smile". In our work, we design an attack on the latent encoding, in previous work (Kos et al., 2017), they perform an attack on the input image.

## 3 METHOD

In this section, we describe how we train a VAE and how we learn the adversarial transform that we apply to the latent encoding.

### 3.1 PROBLEM SETUP

Consider a dataset, $\mathcal{D}$ consisting of labeled binary examples, $\{x_i, y_i\}_{i=1}^N$, for $y_i \in \{0, 1\}$. To perform the mappings between data samples, $x$, and corresponding latent samples, $z$, we learn an encoding process, $q_\phi(z|x)$, and a decoding process, $p_\theta(x|z)$, which correspond to an encoding and decoding function $E_\phi(\cdot)$ and $D_\theta(\cdot)$ respectively. $\phi$ and $\theta$, parameterize the encoder and decoder, respectively. Our objective is to learn an *adversarial* transform, $\hat{\mathcal{T}}$ such that $class(x) \neq class(\hat{\mathcal{T}} \circ x)$, where, $\hat{\mathcal{T}}$, is constrained under an $L_p$ norm. Here, $class(\cdot)$ is the *soft* output of a binary classifier. Rather than applying an adversarial transformation (Moosavi-Dezfooli et al., 2016), $\hat{\mathcal{T}}$ directly to the data, $x$, we propose performing the adversarial transform $\mathcal{T}$ on the latent representation, $\mathcal{T} \circ z$.

We learn a transform, $\mathcal{T}$ with $z = E_\phi(x)$ subject to $class(D_\phi(\mathcal{T} \circ z)) \neq class(D_\phi(z))$[1].

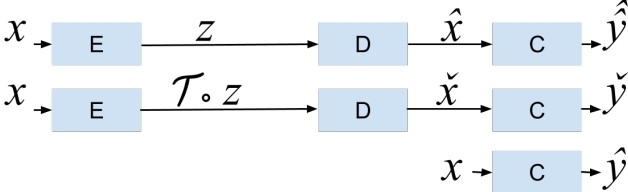

Figure 2: **Notation and Models**: E, D and C are the encoder, decoder and classifier networks.

We consider three methods of attack, and compare two approaches for regularizing $\mathcal{T}$. The three attack methods that we consider are as follows:

1. An *Independent* attack: We consider an attack on a pre-trained variational autoencoder (VAE). $\mathcal{T}$ is learned for the pre-trained VAE.

2. A *Poisoning* attack: We consider an attack during VAE training (poisoning). $\mathcal{T}$ is learned at the same time as the VAE.

3. A *Poisoning+Class* attack: We consider an attack during VAE training, where the VAE is trained not only to reconstruct samples but to produce reconstructions that have low classification error. This, in turn, encourages the VAE to have a discriminative internal representation, possibly making it more vulnerable to attack. We learn $\mathcal{T}$ at the same time.

---

[1]Note than in the case where class labels are binary, this is equivalent to: learning a $\mathcal{T}$ such that $class(D_\phi(\mathcal{T} \circ z)) = 1 - class(D_\phi(z))$.

ADDITIVE PERTURBATION ($z + \Delta z$)

Here, we consider $\mathcal{T} \circ z = z + \Delta z$. There are several options for the form that $\Delta z$ may take. In the first case, $\Delta z$ may be a constant. We may learn a single transform to flip an image with label 0 to an image with label 1, and another for moving in the opposite direction. On the other hand, we may learn a single $\Delta z$ and apply $-\Delta z$ to move in one direction and $+\Delta z$ to move in the other. The advantage of using a constant $\Delta z$ is that at the *attack time* the adversarial perturbation has already been pre-computed, making it easier to attack multiple times. There is a further advantage to using only a single $\Delta z$ because the attacker need only learn a single vector to tamper with (almost) all of the encodings. Alternatively, $\Delta z$ may be a function of any combination of variables $x, y, z$, however, this may require the attacker to learn an attack online – rather than having a precomputed attack that may be deployed easily. In this paper, we are interested in exploring the case where we learn a single, constant $\Delta z$.

We also consider a multiplicative perturbation. However, we reserve explanation of this for the Appendix (Section 7).

## 3.2 LOSS FUNCTIONS

Here, we consider the cost functions used to train a VAE and learn $\mathcal{T}$. The VAE is trained to reconstruct an input, $x$, while also minimizing a Kullback-Leibler ($KL$)-divergence between a chosen prior distribution, $p(z)$ and the distribution of encoded data samples. The parameters of the VAE are learned by minimizing, $J_{vae} = \text{BCE}(x, \hat{x}) + \alpha KL[q_\phi(z|x)||p(z)]$, where BCE is the binary cross-entropy and $\alpha$ is the regularization parameter. A classifier may be learned by minimizing $J_{class} = \text{BCE}(y, \hat{y})$. An additional cost function for training the VAE may be the classification loss on *reconstructed* data samples, $\text{BCE}(y, \hat{y})$. This is similar to an approach used by Chen et al. (Chen et al., 2016) to synthesize class specific data samples. Finally, to learn the *attack transform*, $\mathcal{T}$ we minimize, $J_z = \text{BCE}((1 - y), \check{y}) + \mathcal{L}_p(\mathcal{T})$, for the case above (Section 3.1) we have $\mathcal{L}_p(\mathcal{T}) = ||\Delta z||_p$. This allows us to learn a transform on a latent encoding, that results in a label flip in the decoded image. Minimizing the $L_p$-norm for $p = \{1, 2\}$, encourages the transform to target a minimal number of units of $z$. Specifically, using $p = 1$ should encourage the perturbation vector to be sparse (Donoho, 2006). When $\Delta z$ is sparse, this means that only a few elements of $z$ may be changed. Such minimal perturbations reduce the likelihood that the attack is detected.

## 3.3 EVALUATION METHOD

The goal for the attacker is to tamper with the encoding such that the label of the decoded sample is flipped. For example, if the label was 1 initially, following a successful attack, the label should be 0. Rather than assigning binary labels to samples, our classifier outputs values between $[0, 1]$ where 0 or 1 suggests that the classifier is highly certain that a data sample belongs to either class 0 or class 1, while a classifier output of 0.5 means that the classifier is unsure which class the sample belongs to. When an attack is successful, we expect a classifier to predict the class of the reconstructed image with high certainty. Further, for an attack to be undetectable, we would expect a classifier to predict the label of a reconstructed, un-tampered data sample with almost the same certainty as a tampered one. Formally, we may evaluate the quality of an attack by measuring $|\epsilon|$ such that [2]:

$$class(x) = 1 - class(\hat{\mathcal{T}} \circ x) + \epsilon$$
$$class(D_\theta(z)) = 1 - class(D_\theta(\mathcal{T} \circ z)) + \epsilon$$

Based purely on the classification loss, in the case where $\epsilon = 0$, the encodings that have been tampered with would be indistinguishable from those that had not. An attack may be considered undetectable if $|\epsilon|$ is small. Typically, $|\epsilon|$ may be related to the standard deviation in classification results.

To calculate epsilon we make two practical alterations. The first is that our classifier outputs values $[0, 1]$, which do not necessarily correspond to probabilities, but may in some respect capture the *confidence* of a single classification. Using the output of the classifier, we compute confidence

---

[2] We assume $class(x) = class(\hat{x})$.

scores, where $0$ corresponds to low confidence and $1$ to high confidence. For a sample whose true label is $1$, the confidence is taken to be the output of the classifier. For a sample whose true label is $0$, the confidence is taken to be $(1 - class(\cdot))$, where $class(\cdot)$ is the output of the classifier. The second, is that if the classifier is more confident when classifying one class compared to the other, it does not make sense to compare $class(x)$ to $class(\hat{\mathcal{T}} \circ x)$. Rather, we compare:

$$class(x^{(y=1)}) = class(\hat{\mathcal{T}} \circ x^{(y=0)}) + \epsilon$$

$$class(D_\theta(z^{(y=1)})) = class(D_\theta(\mathcal{T} \circ z^{(y=0)})) + \epsilon$$

where $x^{y=0}$ and $x^{y=1}$ are a data samples with true labels $0$ and $1$ respectively. $z^{y=0}$ and $z^{y=1}$ are encodings of data samples $x^{y=0}$ and $x^{y=1}$, respectively.

We measure the performance of all attacks using the same classifier, so that we may compare attack types more easily. As a consequence, we are also able to show that the attack is partially agnostic to the classifier, provided that the classifier is trained to perform a similar task.

We discuss an additional probabilistic evaluation method in Section 6.4 of the Appendix.

## 4 EXPERIMENTS AND RESULTS

We compare 3 methods of attack using 2 different types of regularization on $\Delta z$ – totaling 6 experiments. The three methods of attack are listed in Section 3 and the two types of regularization are the $L_1$-norm and the $L_2$-norm. We show qualitative results for only two examples in the main text and reserve the rest for the appendix. We provide a quantitative analysis in the form of confidence score (discussed in Section 3.3) for all 6 attack types.

### 4.1 DATASET

Experiments are performed on the CelebA dataset consisting of 200k colour images of faces, of which 100 are reserved for testing. The samples are of size $64 \times 64$, and we do not crop the images. Each image is assigned a binary label, $1$ for smiling and $0$ for not smiling.

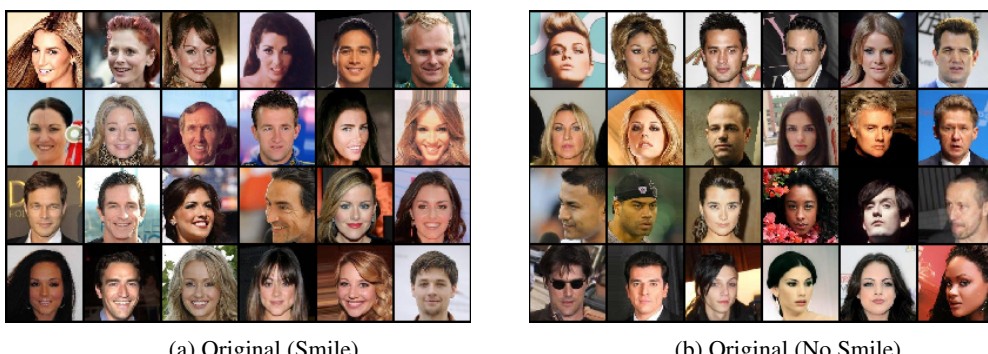

(a) Original (Smile)            (b) Original (No Smile)

Figure 3: **The dataset.** Snippet of (test)images with two labels – smiling and non-smiling.

### 4.2 USING $(z + \Delta z)$ WITH $L_2$ REGULARIZATION

In this section, we focus on adversaries that have been trained using $L_2$ regularization. Figure 4 shows the results of an adversarial attack, where the adversary is learned for a pre-trained VAE, which was trained without label information. We expected this to be a more challenging form of attack since the VAE would not have been trained with any discriminative label information – making it less likely to learn features specifically for "smile" and "not smile". Visual examples of decoded tampered and non-tampered encodings are shown in Figure 4. Figure 4(a) shows reconstructed images of people smiling, while (b) shows similar faces, but without smiles (attacked). Similarly, Figure 4(c) shows reconstructed images of people that are not smiling, while (d) shows similar faces smiling (attacked). In most cases, the success of the attack is obvious.

Quantitative results in Table 1 show several important results. In all cases, the decoded tampered-encodings are classified with high confidence. This is higher than the classifier on either the original image or the reconstructed ones. This suggests that the adversarial attack is successful as tampering with the encoding. By only evaluating the attacks by the confidence, it appears that all adversaries perform similarly well for all attack types. However, it is important to consider the difference between the confidence of reconstructed samples and the samples whose encoding was tampered with. Since the attacker aims to directly optimize the classification score, it is no surprise that affected samples have higher confidence score. It does, however, make the attack potentially more detectable. From this perspective, the more successful attacks would be those whose difference between confidence scores is small (see Section 3.3).

For this particular set of attacks, the most *stealthy* would be switching from "no smile" to "smile" attacking a VAE trained using label information. We may expect a VAE trained with label information to be a particularly good target as it is already trained to learn discriminative features. We also notice that it is easier for the attacker to move in the direction from "no smile" to "smile" than the reverse. The reason for this may be related to the slight bias in the classification results. However, this may also stem from the subjective labelling problem. Some of the faces in Figure (a) that belong to the "smile" class are not clearly smiling.

Both the qualitative results in Figure 4 and the quantitative results in Table 1 indicate successful attack strategies. Further, visual results are shown in the Appendix for the other attack methods, and images showing the pixel-wise difference between reconstructions and attacked samples are also shown (Figure 11) to highlight the effects of $\mathcal{T}$.

Table 1: Confidence scores ($p = 2$) for additive perturbation attacks of types: Independent, Poisoning, Poisoning+Class.

| Data Samples | Independent | Poisoning | Poisoning+Class |
|---|---|---|---|
| Original smile | 0.80 | 0.80 | 0.80 |
| Smile reconstruction | 0.79 | 0.88 | 0.86 |
| No smile $\rightarrow$ Smile | 0.98 | 0.98 | 0.97 |
| Original no smile | 0.93 | 0.93 | 0.93 |
| No smile reconstruction | 0.85 | 0.87 | 0.96 |
| Smile $\rightarrow$ No smile | 0.95 | 0.96 | 0.96 |

## 4.3 USING $(z + \Delta z)$ WITH $L_1$ REGULARIZATION

In this section, we look at results for attacks using $L_1$ regularization on the encoding. $L_1$ regularization is intended to encourage sparsity in $\Delta z$, targeting only a few units of the encoding. In Figure 10 in the appendix, we show that $L_1$ regularization does indeed lead to a more sparse $\Delta z$ being learned.

In Figure 5, we show visual results of an adversarial attack, with the original reconstructions on the left and the reconstructions for tampered encodings on the right. We show examples of all 3 types of attack, with $L_1$ regularization in the appendix. The attack appears to be successful in all cases. We visualize the pixel-wise change between reconstructions of encodings and tampered encodings in Figure 11 of the appendix. Note that our results are not "cherry picked", but simply chosen randomly.

Table 2 shows confidence values for each type of attack when using $L_1$ regularization on $\Delta z$. In all cases, the confidence values for the samples which were attacked is higher than both reconstructed samples and original data samples. This is likely to be because the adversary is picking a perturbation that directly optimises the classification score. It is, however, important to remember that the classifier used to evaluate the attack is the same for all attacks and not the same one used for training the adversary.

As before, if there is a clear difference in confidence score between the reconstructed data samples and the decoded tampered-encodings, it will be obvious that an attack has taken place. If we consider the difference between these scores, then the most *stealthy* attacks are those learning the $\Delta z$ at the

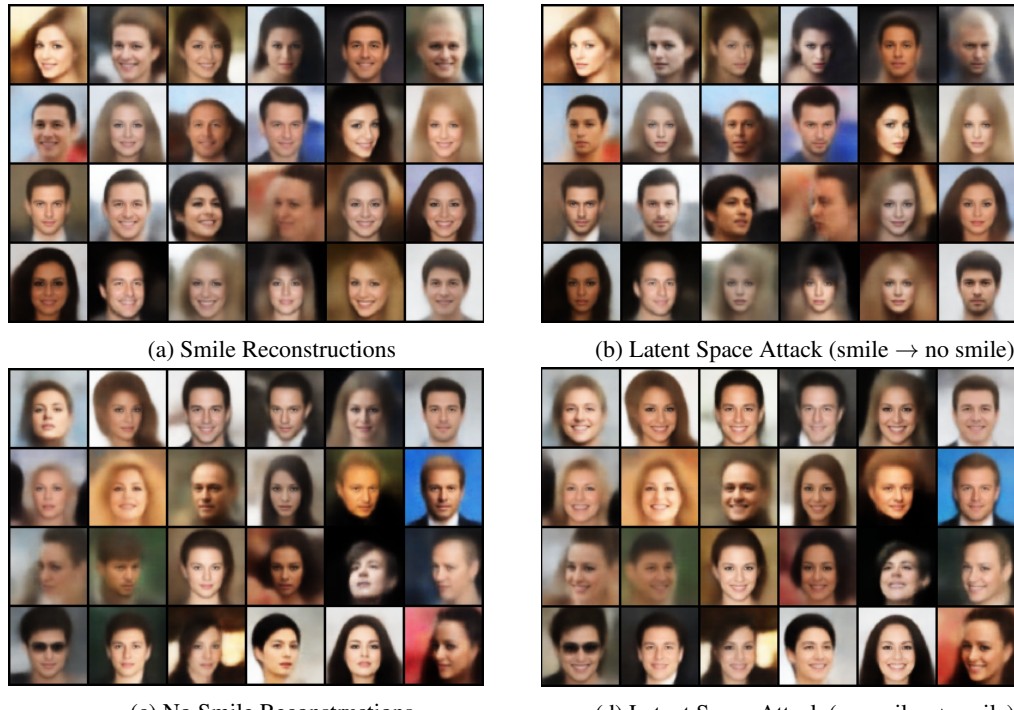

(a) Smile Reconstructions
(b) Latent Space Attack (smile → no smile)

(c) No Smile Reconstructions
(d) Latent Space Attack (no smile → smile)

Figure 4: **Under $L_2$ regularization** Random examples of decoded latents with and without additive perturbation in an Independent attack. Smile (a) and (d), no smile (b) and (c).

Table 2: Confidence scores (p=1) for additive perturbation attacks of types: Independent, Poisoning, Poisoning+Class

| Data Samples | Independent | Poisoning | Poisoning+Class |
|---|---|---|---|
| Original smile | 0.80 | 0.80 | 0.80 |
| Smile reconstruction | 0.87 | 0.78 | 0.80 |
| No smile → smile | 0.94 | 0.98 | 0.98 |
| Original no smile | 0.93 | 0.93 | 0.93 |
| No smile reconstruction | 0.87 | 0.91 | 0.89 |
| Smile → No smile | 0.96 | 0.91 | 0.96 |

same time as learning the VAE to switch between "no smile" and "smile". Similarly, with the results obtained with $L_2$ regularization on $\Delta z$, the more successful attack – in terms of stealth – is to go from "no smile" to "smile" for all attack types.

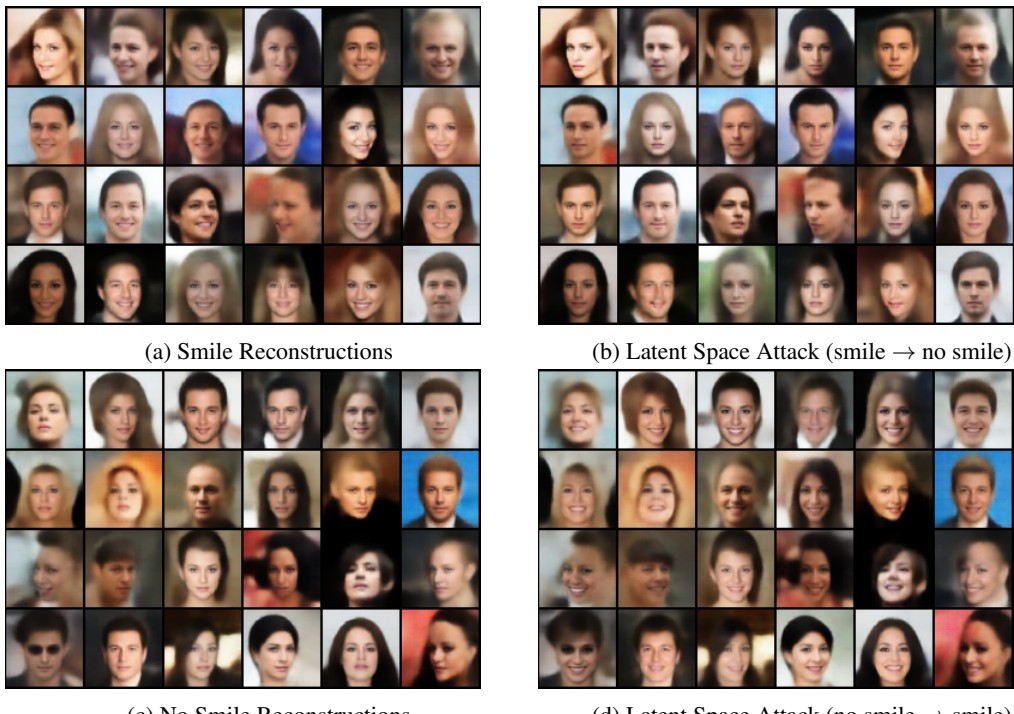

|  |  |
|---|---|
| (a) Smile Reconstructions | (b) Latent Space Attack (smile $\rightarrow$ no smile) |
| (c) No Smile Reconstructions | (d) Latent Space Attack (no smile $\rightarrow$ smile) |

Figure 5: **Under $L_1$ regularization.** Random examples of decoded latents with and without additive perturbation in a Poison+Class attack. Smile (a) and (d), no smile (b) and (c).

## 5    DISCUSSION AND CONCLUSION

In this paper, we propose the idea of latent poisoning – an efficient methodology for an adversarial attack i.e., by structured modification of the latent space of a variational autoencoder. Both additive and multiplicative perturbation, with sparse and dense structure, show that it is indeed possible to flip the predictive class with minimum changes to the latent code.

Our experiments show that additive perturbations are easier to operationalize than the multiplicative transformation of the latent space. It is likely that additive perturbations have reasonable performance because of the near-linear structure of the latent space. It has been shown that given two images and their corresponding points in latent space, it is possible to linearly interpolate between samples in latent space to synthesize intermediate images that transit smoothly between the two initial images (Kingma & Welling, 2013; Radford et al., 2015). If the two images were drawn from each of the binary classes, and a smooth interpolation existed between them, this would mean that additive perturbation in the latent space, along this vector, would allow movement of samples from one class to the other.

How can we counter such a poisoning of the latent space? It might be helpful to look into the predictive probability and its uncertainty on outputs from an autoencoder. If the uncertainty is above a threshold value, an attack may be detected. Detection via predictive probability and its uncertainty, as well as alternative methods, such as inspection of the latent encoding, become even more difficult when the attacker has altered the latent distribution minimally (under a norm).

Given the prevalence of machine learning algorithms, the robustness of such algorithms is increasingly becoming important (McDaniel et al., 2016; Abadi et al., 2017), possibly at par with reporting test error of such systems.

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

## 6 Appendix

### 6.1 Samples with and with out label switch

In the main body of the text, we showed received images for the case where an attack has taken place for two types of attack. In this section, we show the remaining examples.

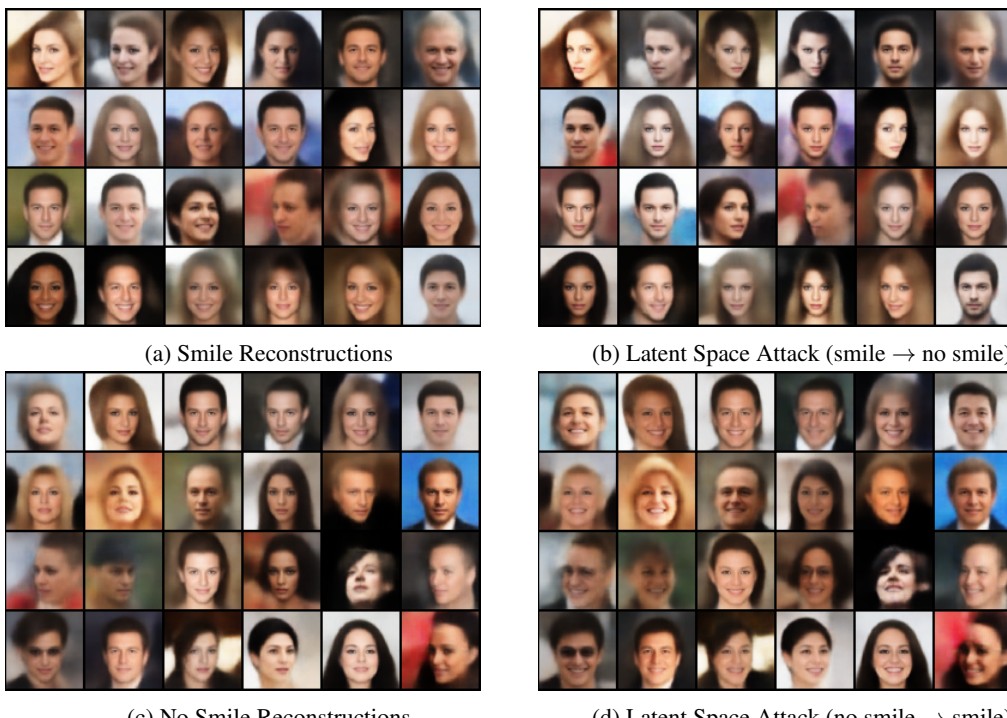

(a) Smile Reconstructions

(b) Latent Space Attack (smile → no smile)

(c) No Smile Reconstructions

(d) Latent Space Attack (no smile → smile)

Figure 6: **Under $L_1$ regularization.** Random examples of decoded latents with and without additive perturbation in an Independent attack. Smile (a) and (d), no smile (b) and (c).

## 6.2 COMPARE USING $|\Delta z|_1$ WITH $|\Delta z|_2$

In this section, we compose Tables of values and figures to compare the 3 different attacks for the 2 different regularization methods.

## 6.3 ENTROPY OF PERTURBATION

We expect that using $L_1$ regularization will give more sparse perturbations, $\Delta z$ than using $L_2$ regularization. In Figure 10, we show the effect of the regularization term for each attack type: (1) learning a $\Delta z$ for a pre-trained VAE, (2) learning a $\Delta z$ while training a VAE and (3) learning a $\Delta z$ while training a VAE and using class information to train the VAE. It is clear from Figure 10 that using $L_1$ regularization does indeed result in a more sparse $\Delta z$.

## 6.4 CAN WE USE KNOWLEDGE OF THE PRIOR TO DETECT AN ADVERSARIAL ATTACK?

Figure 10 provides information about the magnitude of the adversarial perturbations. Here, we consider how knowledge of the magnitude of the perturbations, may allow us to understand the probability of an attack being detected. We consider an approach to individually test each element of a latent encoding to see if we can determine whether an attack has taken place. We refer to a single element of the perturbation $\Delta z$, as $\delta z$ and consider whether we can detect perturbation to a single element in isolation from the other elements in the encoding.

In a variational autoencoder, the distribution of encoded data samples is trained to belong to a chosen prior distribution – in this case a Gaussian. Assuming that the autoencoder is trained well, we may say that the distribution of encoded data samples is Gaussian. Further, we assume that each element in the encoding is drawn independently from the Gaussian distribution. From this, we know that $c.99.5\%$ each individual encoding value lies between $-2.807\sigma$ and $2.807\sigma$ where sigma is the

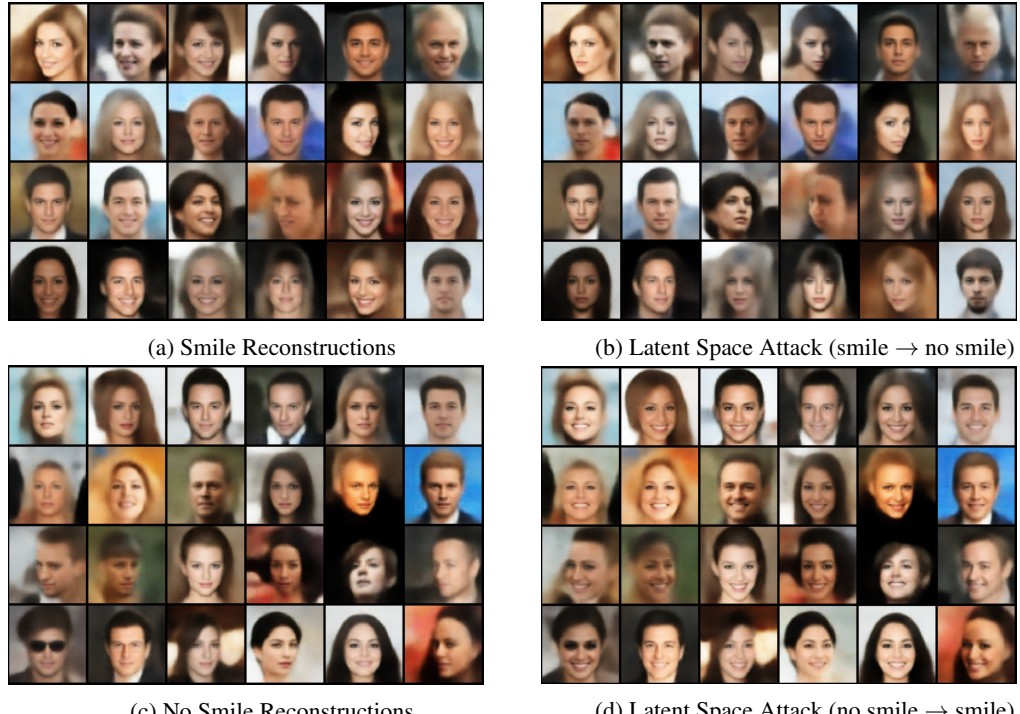

(a) Smile Reconstructions  (b) Latent Space Attack (smile → no smile)

(c) No Smile Reconstructions  (d) Latent Space Attack (no smile → smile)

Figure 7: **Under $L_2$ regularization.** Random examples of decoded latents with and without additive perturbation in a poisoning attack. Smile (a) and (d), no smile (b) and (c).

standard deviation of the Gaussian distribution. This means that approximately $1/200$ [3] elements lie outside this interval. In our case $\sigma = 1$.

Any addition to samples from Gaussian distribution results in a shift of the distribution. For an adversarial attack involving the additive perturbation of $\delta z$ on a single unit of $\Delta z$, we may calculate the probability that a single element in a tampered encoding lies outside the range $[-2.807, 2.807]$. The formula for this is given by:

$$P_{99.5\%}(\delta z) = 1 - \frac{1}{2}\left[1 + \text{erf}\left(\frac{2.807 - \delta z}{\sqrt{2}}\right)\right] + \frac{1}{2}\left[1 + \text{erf}\left(\frac{-2.807 - \delta z}{\sqrt{2}}\right)\right]$$

where $\text{erf}(\cdot)$ is the error function. Note that $P_{99.5\%}(1) = 0.04$, $P_{99.5\%}(2) = 0.2$ and $P_{99.5\%}(5) = 0.98$.

We may use this to evaluate our attack processes and may also be used to further regularize our models to ensure that the probability of being detected is less than a chosen threshold. Looking at Figure 10 we can see that only attacks in (a) and (b) using $L_2$ regularization are likely to be undetectable according to the criteria above, assuming that the encoded data samples follow a Gaussian distribution.

## 6.5 THE EPSILON GAP

Here, we compare the $\epsilon$-gap (described in Section 3.3) for each type of attack, using each type of regularization. We expected that using $L_1$ regularization would encourage minimal change to the encoding needed to make a switch between labels. Therefore we might expect this to influence the epsilon value. However, for a sparse $\Delta z$ to have the desired properties we also require the structure of the latent space to be sparse. Since we did not enforce any sparsity constraint on the latent encoding when training the VAE, sparsity on the latent samples is not guaranteed. Therefore,

---

[3]our latent encoding is of size 200, however the choice of a 99.5% is fairly arbitrary and may be chosen more precisely depending on application.

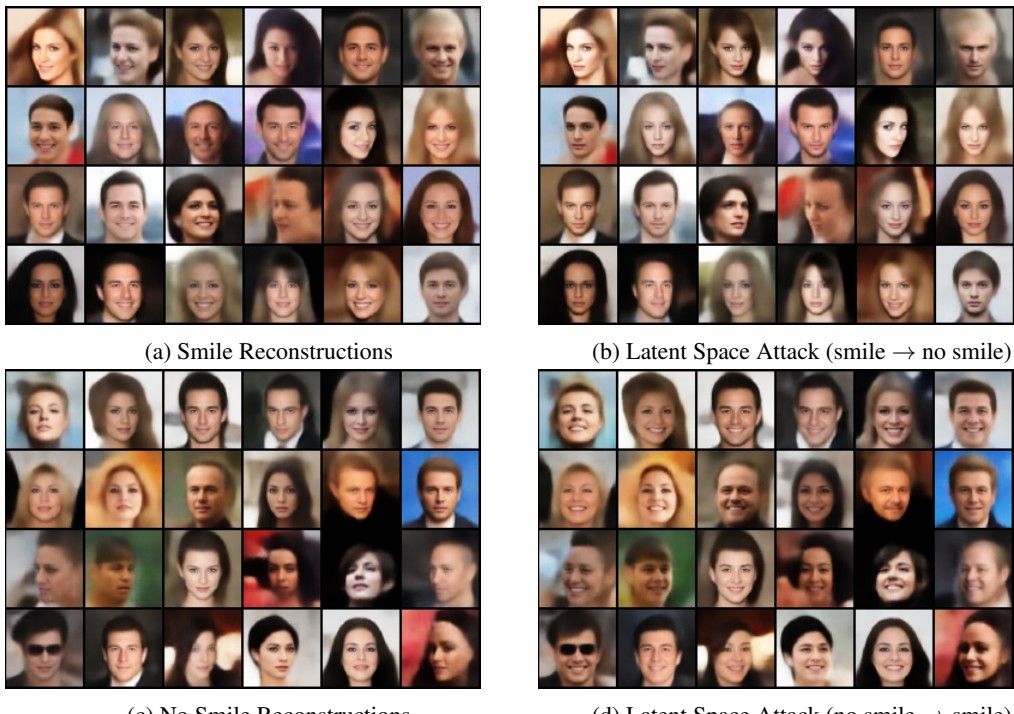

(a) Smile Reconstructions

(b) Latent Space Attack (smile → no smile)

(c) No Smile Reconstructions

(d) Latent Space Attack (no smile → smile)

Figure 8: **Under $L_1$ regularization.** Random examples of decoded latents with and without additive perturbation in an poisoning attack. Smile (a) and (d), no smile (b) and (c).

although it is useful to learn sparse encodings to facilitate the speed of the attack (minimal number of changes to the encoding), it does not clearly affect the overall quality of the attack.

Table 3: Epsilon gap values

| Samples | $-\Delta z$ | | $+\Delta z$ | |
|---|---|---|---|---|
| | p=1 | p=2 | p=1 | p=2 |
| Learn $\Delta z$ & Independent | 0.07 | 0.19 | 0.09 | 0.10 |
| Learn $\Delta z$ & Poisoning jointly | 0.20 | 0.10 | 0.00 | 0.09 |
| Learn $\Delta z$ & Poisoning+Class | 0.18 | 0.11 | 0.07 | 0.00 |

## 6.6 THE EFFECT OF $\Delta z$ ON $\Delta x$

In Figure 11 we show the difference between the reconstructed data samples and decoded tampered-encodings. These images highlight the effect of the adversarial perturbation – applied to the latent space – in the data space.

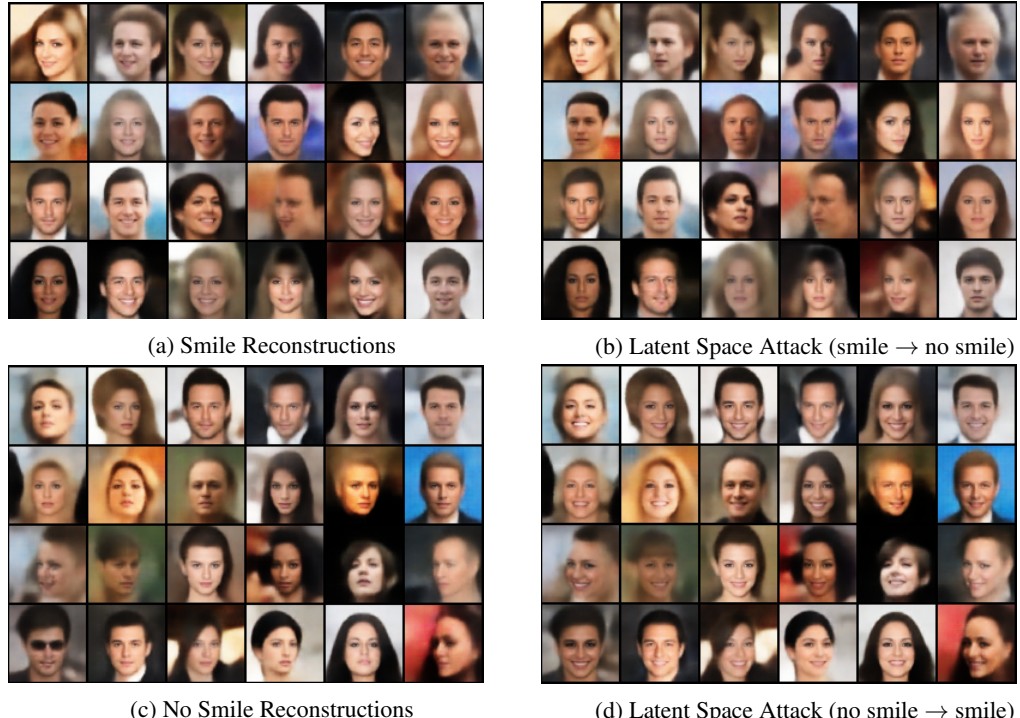

(a) Smile Reconstructions

(b) Latent Space Attack (smile → no smile)

(c) No Smile Reconstructions

(d) Latent Space Attack (no smile → smile)

Figure 9: **Under $L_2$ regularization.** Random examples of decoded latents with and without additive perturbation in an poisoning+Class attack. Smile (a) and (d), no smile (b) and (c).

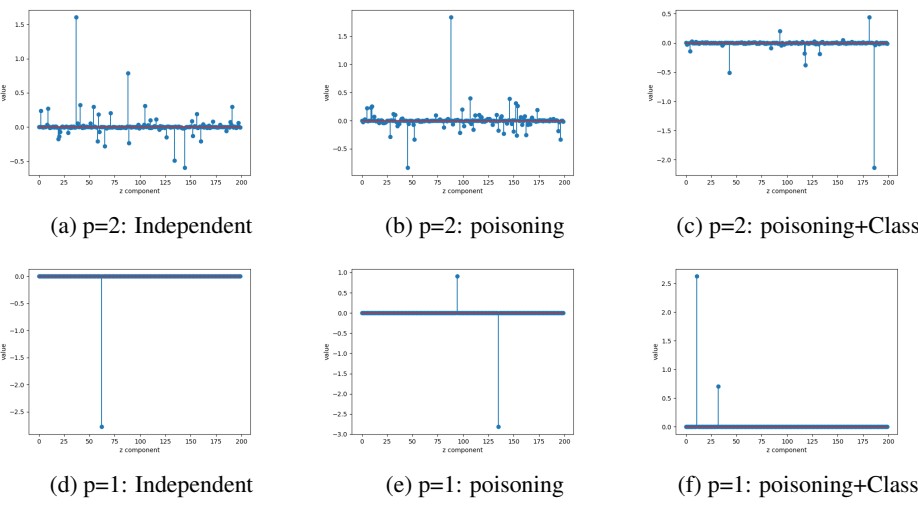

(a) p=2: Independent

(b) p=2: poisoning

(c) p=2: poisoning+Class

(d) p=1: Independent

(e) p=1: poisoning

(f) p=1: poisoning+Class

Figure 10: Visualization of the values of each element in the learned $\Delta z$ in an additive perturbation attack. The $x$-axis corresponds to units in $\Delta z$ and the $y$-axis to the values that each unit takes. This figure demonstrates the effect of $L_1$ and $L_2$ regularization on the sparsity of the $\Delta z$ learned.

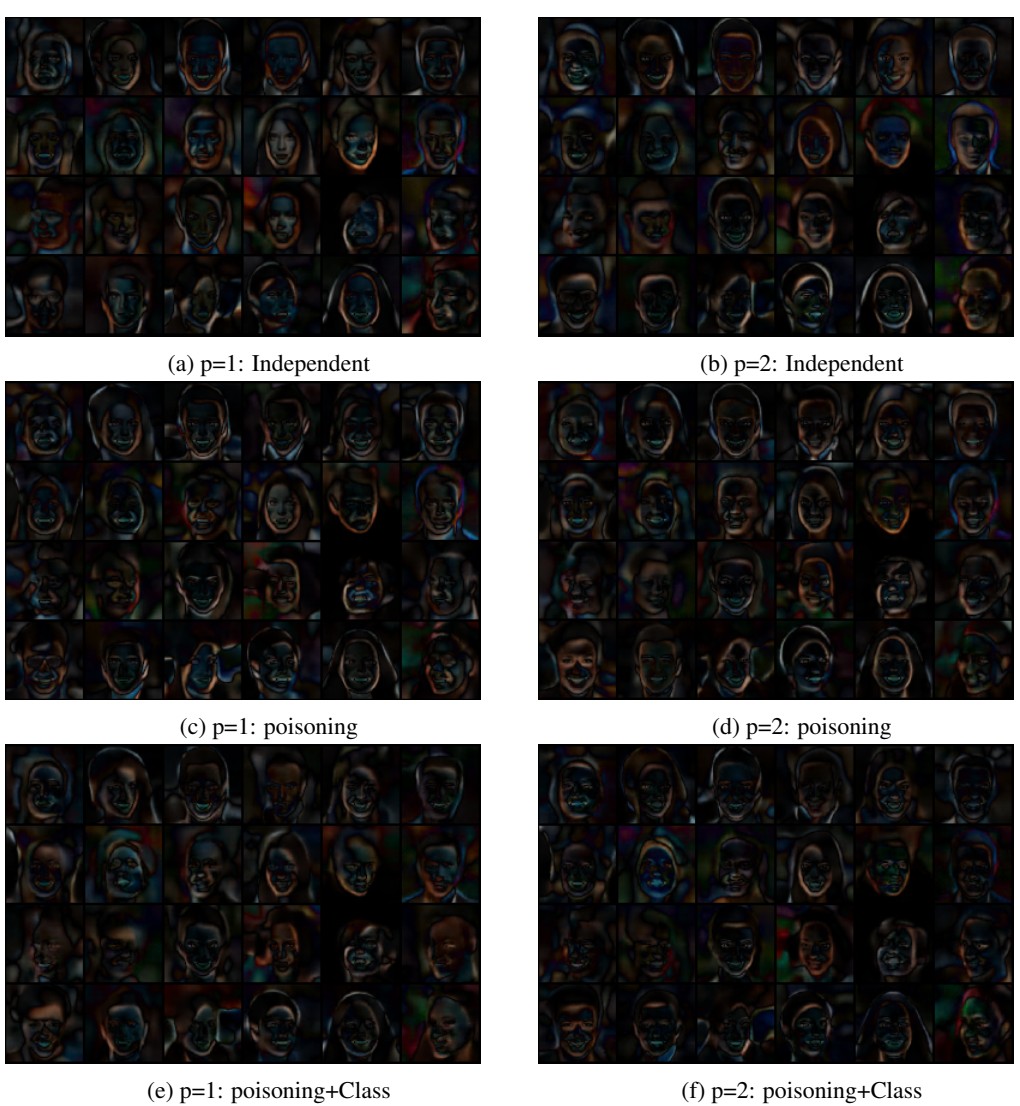

(a) p=1: Independent

(b) p=2: Independent

(c) p=1: poisoning

(d) p=2: poisoning

(e) p=1: poisoning+Class

(f) p=2: poisoning+Class

Figure 11: Difference image between decoded encodings (reconstructions) and decoded tampered-encodings for additive perturbation attacks. Here we show only results in the direction of $+\Delta z$.

## 7 IMPLEMENTATION DETAILS

For both the encoder, decoder and classifier we use an architecture similar to that used by Radford et al. Radford et al. (2015). We weight the $KL$-divergence in the VAE loss by $\alpha = 0.1$ and we train the model using Adam with a learning rate of $2e - 4$ however training was not sensitive to this parameter – training with a learning rate of $1e - 3$ also worked. Our code both for training (with all parameter values) and evaluation will be made available after the review process via Github.

MULTIPLICATIVE PERTURBATION $z \cdot (1 + \Delta z)$

To formulate a multiplicative perturbation, we require that the element(s) that encode smile or no smile have different signs for each class. We may then learn a multiplicative mask, where most of the values are ones, and one or a few values are negative. The values may not be positive. If the values are positive then signs in the encoding cannot be switched and no label swap may take place. In this formulation, we cannot guarantee that the encoding will take the desired form. From preliminary experiments, we see that faces classified as "smiling" often appear to be smiling more intensely after the transform. This is likely to be because the autoencoder considered the image to be a person not smiling in the first place.

In our formulation, we use a single $\Delta z$ to which we apply $L_p$ regularization to. The transform is then $z(1 + \Delta z)$. Note that it does not make sense to have a formulation for each direction i.e. $z(1 - \Delta z)$ for the other direction; if the encoding for opposite samples has opposite signs a negative $\Delta z$ is sufficient to provide a transform in both directions.

For multiplicative transforms, the perturbations do not appear to perform as well as for the additive approach. This might be a reflection of the near-linear structure of the latent space learned by the autoencoder. An adversary applying an additive perturbation is able to target the near-linear structure, while an adversary applying a multiplicative perturbation implies much stronger assumptions on the structure of the latent space – which apparently do not hold for all variational autoencoders.

