# OpenReview forum: "LatentPoison -- Adversarial Attacks On The Latent Space"
_ICLR.cc/2018/Conference — Reject_

### Official Review · AnonReviewer2 · 2017-11-27
**LatentPoison -- Adversarial Attacks On The Latent Space**

**Rating:** 5
**Confidence:** 3

**Review:**

The idea is clearly stated (but lacks some details) and I enjoyed reading the paper.

I understand the difference between [Kos+17] and the proposed scheme but I could not understand in which situation the proposed scheme works better. From the adversary's standpoint, it would be easier to manipulate inputs than latent variables. On the other hand, I agree that sample-independent perturbation is much more practical than sample-dependent perturbation.

In Section 3.1, the attack methods #2 and #3 should be detailed more. I could not imagine how VAE and T are trained simultaneously.

In Section 3.2, the authors listed a couple of loss functions. How were these loss functions are combined? The final optimization problem that is used for training of the propose VAE should be formally defined. Also, the detailed specification of the VAE should be detailed.

From figures in Figure 4 and Figure 5, I could see that the proposed scheme performs successfully in a qualitative manner, however, it is difficult to evaluate the proposed scheme qualitatively without comparisons with baselines. For example, can the proposed scheme can be compared with [Kos+17] or some other sample-dependent attacks? Also, can you experimentally show that attacks on latent variables are more powerful than attacks on inputs?

---

> ### Author Response · Authors · 2018-01-04
> **Response to AnonReviewer2**
>
> Thank you for your careful reading of our paper and for providing useful feedback. We have addressed your comments below:
>
> 1. I understand the difference between [Kos+17] and the proposed scheme but I could not understand in which situation the proposed scheme works better.
> (a) From the adversary's standpoint, it would be easier to manipulate inputs than latent variables.
> (b) On the other hand, I agree that sample-independent perturbation is much more practical than sample-dependent perturbation.
>
> Response: (a) If it is possible to perturb the input to the encoder, it makes sense that it should be equally easy to perturb the input to the decoder. Possible confusion here may arise from the fact that the perturbation applied in [Kos2017] is applied to an image selected by the user therefore perturbation must still be applied in an online manner, unlike with traditional adversarial examples {CITE}, that may be synthesised offline and supplied to an algorithm.
>
> 2. In Section 3.1, the attack methods #2 and #3 should be detailed more. I could not imagine how VAE and T are trained simultaneously.
>
> Response: The way that we trained these was, in one epoch to update the parameters of the VAE and then the adversary, T. If you are suggesting that the adversary would not have access to the model during training, we point towards this quotation:
>
> “Attackers may also act during the learning process, for example tampering with some of the training data, or reading intermediate states of the learning system.” - Abadi et al. (On the Protection of Private Information in Machine Learning Systems, 2017).
>
> 3. In Section 3.2, the authors listed a couple of loss functions. How were these loss functions are combined? The final optimization problem that is used for training of the propose VAE should be formally defined. Also, the detailed specification of the VAE should be detailed.
>
> Response: In section 3.2 we define 3 losses: Jvae for updating the VAE, Jclass for updating the classifier for method #3 only and Jz for updating the adversarial transform model, T. The loss functions were not combined because they are each used to train a different model in the system.
> (a) In method #1 the VAE and T are not updated at the same time therefore it does not make sense to combine Jvae and Jz in the same cost function. The VAE is first trained to minimise Jvae, then T is trained to minimise Jz. The classifier is learned separately too, and does not update the parameters of the VAE.
> (b) In method #2, for each epoch the VAE is updated using Jvae and then T is updated using Jz. The losses update different sets of parameters. The classifier is learned separately and does not update the parameters of the VAE.
> (c) In method #3, arguably we could have written a new cost function, J = Jvae + Jclass, however since this only applied to method #3 we did not include this.
>
> 4. From figures in Figure 4 and Figure 5, I could see that the proposed scheme performs successfully in a qualitative manner, however, it is difficult to evaluate the proposed scheme qualitatively without comparisons with baselines. For example, can the proposed scheme can be compared with [Kos+17] or some other sample-dependent attacks? Also, can you experimentally show that attacks on latent variables are more powerful than attacks on inputs?
>
> These experiments could certainly be carried out. However, there would not be space to add this comparison here.

---

### Official Review · AnonReviewer1 · 2017-11-27
**VAE are not a compression scheme**

**Rating:** 3
**Confidence:** 4

**Review:**

This paper misses the point of what VAEs (or GANs, in general) are used for. The idea of using VAEs is not to encode and decode images (or in general any input), but to recover the generating process that created those images so we have an unlimited source of samples. The use of these techniques for compressing is still unclear and their quality today is too low. So the attack that the authors are proposing does not make sense and my take is that we should see significant changes before they can make sense.

But let’s assume that at some point they can be used as the authors propose. In which one person encodes an image, send the latent variable to a friend, but a foe intercepts it on the way and tampers with it so the receiver recovers the wrong image without knowing. Now if the sender believes the sample can be tampered with, if the sender codes z with his private key would not make the attack useless? I think this will make the first attack useless.

The other two attacks require that the foe is inserted in the middle of the training of the VAE. This is even less doable, because the encoder and decoder are not train remotely. They are train of the same machine or cluster in a controlled manner by the person that would use the system. Once it is train it will give away the decoder and keep the encoder for sending information.

---

> ### Author Response · Authors · 2018-01-04
> **Response to AnonReviewer1**
>
> Thank you for your comments, we have addressed them below:
>
> 1. This paper misses the point of what VAEs (or GANs, in general) are used for. The idea of using VAEs is not to encode and decode images (or in general any input), but to recover the generating process that created those images so we have an unlimited source of samples. The use of these techniques for compressing is still unclear and their quality today is too low. So the attack that the authors are proposing does not make sense and my take is that we should see significant changes before they can make sense.
>
> Response: While we appreciate that the specific VAE architecture is not directly used for image compression, autoencoders more generally have been used for image compression {Theis2017} and outperform industry standard compression techniques such as JPEG. We choose to look at the VAE as an example of *an* autoencoder as this appeared to be a good starting point for research in this area. Further, it was not possible (given the space and resource constraints) to apply this technique to all state of art autoencoding compression models. Instead, we chose a model that shares many properties with state-of-art autoencoding compression models.
>
> Deep autoencoders are being developed for tasks such as compression and the purpose of our paper is to expose vulnerabilities in deep autoencoders, so that these vulnerabilities may be kept in mind when developing such algorithms.
>
> 2. But let’s assume that at some point they can be used as the authors propose. In which one person encodes an image, send the latent variable to a friend, but a foe intercepts it on the way and tampers with it so the receiver recovers the wrong image without knowing. Now if the sender believes the sample can be tampered with, if the sender codes z with his private key would not make the attack useless? I think this will make the first attack useless.
>
> Response: If the sender encodes z with his own private key, rather than with the encoder, the decoder will not be able to decode the image at all.
>
> 3. The other two attacks require that the foe is inserted in the middle of the training of the VAE. This is even less doable, because the encoder and decoder are not train remotely. They are train of the same machine or cluster in a controlled manner by the person that would use the system. Once it is train it will give away the decoder and keep the encoder for sending information.
>
> Response: The adversary itself does not affect the training. It simply learns along side the autoencoding model. The only assumption we make is that the adversary can access the encodings and see the outputs of the autoencoder model during training. Abadi et al. suggest that this is not an unreasonable assumption:
>
> “Attackers may also act during the learning process, for example tampering with some of the training data, or reading intermediate states of the learning system.” - Abadi et al. (On the Protection of Private Information in Machine Learning Systems, 2017).
>
> Further, training on clusters often suggests training remotely and transfer of models, data and results between machines.

---

### Official Review · AnonReviewer3 · 2017-11-27
**Too simplistic scenario**

**Rating:** 4
**Confidence:** 4

**Review:**

This paper is concerned with both security and machine learning.
Assuming that data is encoded, transmited, and decoded using a VAE,
the paper proposes a man-in-middle attack that alters the VAE encoding of the input data so that the decoded output will be misclassified.
The objectives are to: 1) fool the autoencoder; the classification output of the autoencoder is different from the actual class of the input; 2) make minimal change in the middle so that the attack is not detectable.

This paper is concerned with both security and machine learning, but there is no clear contributions to either field. From the machine learning perspective, the proposed "attacking" method is standard without any technical novelty. From the security perspective, the scenarios are too simplistic. The encoding-decoding mechanism being attacked is too simple without any security enhancement. This is an unrealistic scenario. For applications with security concerns, there should have been methods to guard against man-in-the-middle attack, and the paper should have at least considered some of them. Without considering the state-of-the-art security defending mechanism, it is difficult to judge the contribution of the paper to the security community.

I am not a security expert, but I doubt that the proposed method are formulated based on well founded security concepts and ideas. For example, what are the necessary and sufficient conditions for an attacking method to be undetectable? Are the criteria about the magnitude of epsilon given on Section 3.3. necessary and sufficient? Is there any reference for them? Why do we require the correspondence between the classification confidence of tranformed and original data? Would it be enough to match the DISTRIBUTION of the confidence?

---

> ### Author Response · Authors · 2018-01-04
> **Response to AnonReviewer3:**
>
> Thank you for your comments and feedback. We have addressed your concerns below:
>
> 1. This paper is concerned with both security and machine learning, but there is no clear contributions to either field. From the machine learning perspective, the proposed "attacking" method is standard without any technical novelty.
>
> All previous work, to our knowledge, has considered learning attacks on *image* space, not encoding space. Further, the perturbation in image space is such that a valid image can be constructed (but the class is flipped, according to a classifier). Please note that the most similar relevant work of Kos (2017), the attack requires access to the encoder; ours does not.  We consider these to be worthy and significant technical contributions.
>
> 2. From the security perspective, the scenarios are too simplistic. The encoding-decoding mechanism being attacked is too simple without any security enhancement. This is an unrealistic scenario. For applications with security concerns, there should have been methods to guard against man-in-the-middle attack, and the paper should have at least considered some of them. Without considering the state-of-the-art security defending mechanism, it is difficult to judge the contribution of the paper to the security community.
>
> Response: The purpose of the paper is to expose vulnerabilities in deep autoencoders. A significant amount of research has been done to explore the application of autoencoders. Perhaps the most significant are the compressive autoencoders {Theis2017}, some of which out perform JPEG compression. If these systems were to be deployed, it would be appropriate to be aware of their vulnerabilities. It makes sense to simultaneously  develop (1) attack, (2) defence and (3) autoencoder technologies so that when autoencoder technologies are deployed, we know that they can be deployed safely. Our experience with  teams using latent spaces to encode images for downstream tasks, is that latent spaces tend to be thought of as being incorruptible.  This is clearly not the case.
>
> None of the other previous, widely cited papers {Goodfellow2014, Kos2017, Papernot2016} consider any other layers of security, such as encryption. We agree that, ultimately, these should be taken into account, but best practice  in cybersecurity involves examining each layer of security *independently*.
>
> 3. I am not a security expert, but I doubt that the proposed method are formulated based on well founded security concepts and ideas. For example, what are the necessary and sufficient conditions for an attacking method to be undetectable?
>
> Response: Good question.  This is what we started to explore in the appendix (Section 6.4). We do not think the answer can be given in anything other than a probabilistic sense. We implicitly think about the scenario in which the only way to detect attack would be through examining the likelihood that  a given unit, of an encoding, lies outside certain confidence intervals of the encoding distribution.  For this paper, we considered a Gaussian prior on each element of latent encoding space; the logical extension to this would be run experiments and develop the principle behind detecting change in at least one unit.
>
> 4. Are the criteria about the magnitude of epsilon given on Section 3.3. necessary and sufficient? Is there any reference for them?
>
> Response: No, we would only be able to make decisions based on probabilistic measures.
>
> 5. Why do we require the correspondence between the classification confidence of tranformed and original data?
>
> Response: For any particular classifier, we may know it’s classification confidence under normal operating conditions. If an attack occurs, to produce a “transformed” data sample, then it is possible that the confidence with which a sample is classified is different to that of “original” data samples that the model was validated on. If this difference is detectable, then we may say that the attack is detectable. If the difference is not detectable, we may not know if an attack has taken place.
>
> Indeed by simply sending the same sample multiple times and comparing classification confidence, you would be able to tell whether any samples had been contaminated.
>
> These experiments are designed to show that the attack is not easily detectable in the image space.
>
> 6. Would it be enough to match the DISTRIBUTION of the confidence?
>
> It is not entirely clear what you mean by “match”. I agree, that it would make sense to divide a distribution of confidence values for a system operating under normal conditions, and then it may be easier to identify if a corrupted samples is an outlier. However, the distribution is likely to be Gaussian, suggesting that to describe the distribution it is sufficient to know the mean and standard deviation. However, rather than doing this in image space, based on classification confidence, we proposed to do this in latent space. We explore these ideas in the Appendix section 6.4.

---

### Decision · Program_Chairs · 2018-01-29
**ICLR 2018 Conference Acceptance Decision**

**Decision:**

Reject

**Comment:**

The paper proposes to launch adversarial attacks in the latent space of VAE such that the minimal change in the latent representation leads to the decoder producing an image with class predictions altered.  Given the pros/cons the paper in its current form falls short of acceptance.

Pros:
Reviewers agree that the paper is well written and easy to follow

Cons:
- The paper lacks novelty and uses standard attacks and defense methodology.
- Reviewers find the attack scenario presented is unrealistic and hence may not useful.
- Experiments lack rigorous comparisons with baselines and it is not clear if the attack in the latent space will be stronger than the attack in the input space.